# Association between age at first alcohol use and heavy episodic drinking: An analysis of Thailand's smoking and alcohol drinking behavior survey 2017

Paithoon Sonthon[1], Narumon Janma[1], Udomsak Saengow[2,3,4] *

1 Faculty of Science and Technology, Phetchabun Rajabhat University, Phetchabun, Thailand, 2 Center of Excellence in Data Science for Health Study, Walailak University, Tha Sala, Nakhon Si Thammarat, Thailand, 3 School of Medicine, Walailak University, Tha Sala, Nakhon Si Thammarat, Thailand, 4 Research Institute for Health Sciences, Walailak University, Tha Sala, Nakhon Si Thammarat, Thailand

* saengow.udomsak@gmail.com

**Data Availability Statement:** The authors obtained the data used in this study from the Center for Alcohol Studies, Thailand (http://cas.or.th/). We have not been granted permission to share the data

## Abstract

According to evidence from developed countries, age at first alcohol use has been identified as a determinant of heavy episodic drinking (HED). This study aimed to investigate the association between age at first alcohol use and HED using data from the Smoking and Drinking Behavior Survey 2017, a Thai nationally representative survey. Binary logistic regression was used to examine the association. This study used data from 23,073 current drinkers in the survey. The survey participants were chosen to represent the Thai population aged 15 years and older. The prevalence of HED and frequent HED among Thai drinkers was 18.6% and 10.1%, respectively. Age at first drinking <20 years was associated with higher odds of HED (adjusted OR, 1.43; 95% CI, 1.26–1.62) and frequent HED (adjusted OR, 1.31; 95% CI, 1.12–1.53) relative to age at first drinking ≥25 years. Regular drinking, drinking at home, and exposure to alcohol advertising increased the odds of HED. Drinking at home was associated with frequent HED. There was a significant interaction between the effect of age at first alcohol use and sex on HED and frequent HED with a stronger effect of age at first alcohol use observed in females. This study provides evidence from a developing country that early onset of alcohol use is associated with HED. Effective measures such as tax and pricing policy should be enforced to delay the onset of drinking.

## 1. Introduction

Alcohol use is associated with a substantial burden on mortality, morbidity, and healthcare costs [1, 2]. Globally, alcohol use was the leading cause of death among females (3.8%) and males (12.2%) aged 15–49 years [2] and disability-adjusted life years among females (2.3%) and males (8.9%) in 2016 [2].

Heavy episodic drinking (HED) or binge drinking, defined as an episode of drinking five or more standard drinks of alcoholic beverages for males or four or more drinks for females in about 2 hours [3], is a high-risk drinking pattern associated with harm to drinkers and others

publicly. For researchers interested in obtaining data, data can be requested from the Center for Alcohol Studies (contact via taksaya.cas@gmail. com or sdarika_t@hotmail.com).

**Funding:** This work was supported by the Center for Alcohol Studies, Thailand, awarded to US [grant number 62-02029-0043] (http://cas.or.th/cas/). This manuscript was partially supported by the New Strategic Research (P2P) project of Walailak University, Thailand, awarded to US (https://www. wu.ac.th/en/). The funders had no role in study design, data collection and analysis, decision to publish, or preparation of the manuscript.

**Competing interests:** The authors have declared that no competing interests exist.

[4]. HED is associated with alcohol-related injuries [5], violence [6], mortality [7], and high healthcare costs [1]. Drinkers' quality of life is negatively associated with HED [8]. The prevalence of HED among drinkers varied between countries, for instance, 28% in the US [9], 38% in Canada [10], 65.1% in France [11], 36% in Australia [12], 13.7% in Singapore [13], and 11.9% in Thailand [14].

Age at first alcohol use is recognized as an important risk factor for HED, alcohol dependence, and alcohol-related consequences [15–19]. An analysis of the 2010 National Survey on Drug Use and Health in the US found that age at first alcohol use of <12, 12–14, and 15–17 years was associated with 3.0-, 2.6-, and 1.9-fold increase in the likelihood of HED compared to initiation of alcohol drinking at 18–24 years [15]. A prospective study of Australian students and parents reported that earlier age at first alcohol use is associated with binge drinking and a higher quantity of alcohol consumed after adjusting for parental and family factors [12]. Another prospective study from Canada found that early alcohol use was associated with current binge drinking and current alcohol use [10]. Other factors including exposure to alcohol advertising [20], drinking contexts [21], smoking [22], income [23], and sex [10] are associated with HED.

Most studies on the association between age at first alcohol use and HED have been conducted in developed countries where alcohol use and binge drinking have decreased since the early 2000s [24–28]. This trend is less obvious in developing countries [24]. In Thailand, the prevalence of current drinkers among individuals aged 15–19 years has slightly increased from 11.0% in 2001 to 13.6% in 2017 [14]. Nevertheless, the prevalence of youth drinking is generally higher in developed countries than in developing countries [29, 30]. Although the initially high prevalence of youth drinking in developed countries has been declining, no obvious decrease was noted in the initially low prevalence in developing countries.

This difference in the prevalence and trend of youth drinking between developed countries and developing countries may reflect the difference in contextual factors that are associated with the trend of youth drinking [24, 28, 31] and the relationship between age at first alcohol use and HED [16]. The association between age at first alcohol use and HED in developing countries may differ from what is observed in developed countries. However, evidence on the association from developing countries is lacking. With the trend of youth drinking observed in some developing countries such as Thailand, knowledge about the association can facilitate decision-making for implementing measures to delay youth drinking. This study aimed to investigate the association between age at first alcohol use and HED using data from a nationally representative survey in Thailand.

To provide context for this study, nearly one-third of Thais are current drinkers. Between 2007 and 2017, the prevalence of current drinkers ranged between 28.4% and 32.3% [14], whereas the prevalence was 59.9% in WHO's European Region and 54.1% in WHO's Region of the Americas in 2016 [29]. This distinction can be attributed in part to Thailand's cultural background. Thailand is a predominantly Buddhist country. Thai people adhere to Buddhism's teaching known as the Five Percepts, which include refraining from intoxication—including alcohol use. The Buddhist Lent Abstinence Campaign, which encourages people to abstain from drinking for three months, is Thailand's major sobriety campaign. In 2016, almost six million Thai drinkers were estimated to abstain completely during the campaign period [32]. Alcohol regulations in Thailand are implemented primarily under the Alcohol Beverage Control Act, B.E. 2551 (2008). The National Alcohol strategy is a blueprint for alcohol policy. It comprises five strategies: controlling economic and physical access to alcohol, altering social norms toward alcohol and reducing drinking motivation, reducing harms from drinking, promoting community-based solutions, and creating policy supporting mechanisms [33].

## 2. Methods

### 2.1 Study design

This study analyzed data from the Smoking and Drinking Behavior Survey 2017 (SADBeS), a nationwide cross-sectional survey in the Thai population. Variables that were relevant to the research questions were selected for the analysis.

### 2.2 Data source

Data for this study were obtained from SADBeS. The National Statistical Office, Thailand, conducted the survey. A stratified two-stage random sampling technique was used in the survey. In the first stage, enumeration areas (a sampling frame for the national census) were randomly selected with the probability proportional to size from 77 strata (the survey employed provinces as strata). In the second stage, households were randomly chosen from the selected enumeration areas. All household members aged 15 years and above who were fluent in Thai were invited to participate in the survey. The response rate of the survey was 93.5%.

The survey included data from non-drinkers, former drinkers, and drinkers. The present study analyzed data from current drinkers aged 15 years and above. Participants with missing data for the required variable were excluded. Therefore, the analysis included only complete cases. Ultimately, this study included data from 23,073 participants. The flow diagram of participant selection is shown in Fig 1.

### 2.3 HED

The outcome of interest was HED. The amount of drinking considered HED differs by sex: five or more standard drinks of alcohol for males and four or more standard drinks for females [3]. The item for HED in the survey followed the definition for males. Hence, male and female participants were asked to respond to the same item for HED. The item was: "How often had

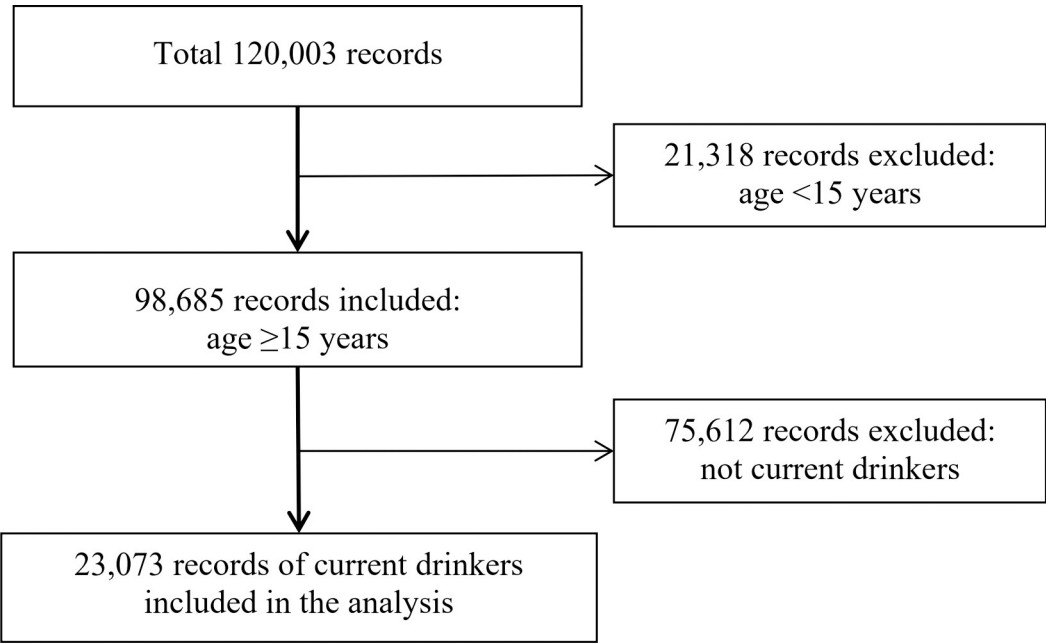

**Fig 1. Participant flowchart.**

you drunk heavily (5 drinks or more) in a short time period in the past 12 months? " The response options were as follows: no, 7 days per week, 5–6 days per week, 3–4 days per week, 1–2 days per week, 1–3 days per month, 8–11 days per year, 4–7 days per year, and 1–3 days per year. Two dichotomous variables were created based on these responses. The first variable, HED, indicated the HED frequency of at least once a month (which corresponds to the option '1–3 days per month' or more frequent). The frequency of at least once a month was chosen to make the estimate somewhat comparable with a 30-day time frame used by the World Health Organization in reporting HED prevalence among drinkers [34]. Nonetheless, as the survey item used a 12-month time frame, the prevalence was underestimated in this analysis. The other variable, frequent HED, indicated the HED frequency of at least once a week (which corresponds to the option '1–2 days per week' or more frequent).

## 2.4 Age at first alcohol use

In this study, the explanatory variable of interest was age at first alcohol use defined as the age at which the individual first drank alcohol. The original question asked in the survey was, "How old were you when you drank alcoholic beverages for the first time?" The response was open-ended. As the legal purchasing age for alcoholic beverages in Thailand is 20 years and above, a new variable was created by categorizing the responses into three categories: <20 years old, 20–24 years old, and ≥25 years old.

## 2.5 Other covariates

Covariates related to demographic characteristics included sex, age, marital status (single, married, and widowed/divorced/separated), education level (primary school or lower, secondary school, and college or higher), household income (<5,000 Thai baht [THB], 5,000–9,999 THB, and ≥10,000 THB), and smoking (no, occasional, and regular). Drinking-related covariates included frequency of drinking (occasional [less frequent than once a week] and regular [once a week of more frequent]), most frequent drinking venue (own home, someone else's home, and party and traditional ceremony venue), and exposure to alcohol advertising (yes, no, and unsure; the unsure response was coded as 'no'). The item for alcohol advertising exposure was a single question, "During the past 30 days, have you seen or heard any alcoholic beverage advertisement?". No further information was given to survey participants regarding alcohol advertising.

## 2.6 Statistical analysis

Percentage, mean, standard deviation, minimum, and maximum were used for describing continuous variables. Categorical variables were represented by count and percentage. The prevalence of HED and frequent HED was computed. Multivariate analysis was performed using a binary logistic regression model. The primary explanatory variable was age at first alcohol use. All other variables included as covariates in the multivariate model were based on existing evidence of association with HED. The dependent variables were HED and frequent HED. Crude and adjusted odds ratios (ORs) with 95% confidence interval (CI) and p-value were estimated using data from the entire sample. As participants aged 24 years or below were unable to be fully coded for the age at first alcohol use variable (e.g., a participant aged 22 years cannot state the age at first alcohol use of ≥25 years old), three separate regression models were run using data from each age group (i.e., 15–19 years, 20–24 years, and 25+ years) to address this issue. All three models had the identical set of covariates as the full sample model. The only difference was how the age at first alcohol use variable was coded. For the youngest age group, a continuous age at first use variable was used in the model. For the 20–24 age

group, a dichotomous age at first use variable (<20 years old and 20–24 years old) was used. The model for the 25+ age group employed the three-level age at first use variable (20 years old, 20–24 years old, and ≥25 years old) as the full sample models.

Since the Thai population's drinking patterns differ considerably between males and females [30], we examined the effect of interaction between age at first alcohol use and sex on HED and frequent HED using a logistic regression model. In all multivariate analyses, only participants with complete data on the outcome variable and covariates were included. All analyses were unweighted and conducted using the R statistical language via RStudio version 4.0.3. A significance level of 0.05 was used for all analyses.

### 2.7 Ethics approval

This study was approved by the Human Research Ethics Committee of Walailak University, Thailand (WU-EC-MD-3-470-63). The data were analyzed anonymously.

## 3. Results

### 3.1 Characteristics of current drinkers

Characteristics of current drinkers are shown in Table 1. The majority were males. The average age was 44.3 years. More than 70.0% of current drinkers were married. Half had primary school education or lower. The mean monthly household income was 10,919.9 THB. The proportion of regular smokers was 38.7%. The average age of first alcohol use was 20.6 years. Almost half of the drinkers drank regularly. One-third of the drinkers were exposed to alcohol advertising. The prevalence of HED and frequent HED was 18.6% and 10.1%, respectively.

### 3.2 Prevalence of HED and frequent HED among drinkers

The prevalence of HED and frequent HED by drinkers' characteristics is shown in Table 2. The prevalence of HED and frequent HED was higher among drinkers with a lower age at first alcohol use. For example, the prevalence of HED was 22.1% among those who began drinking before the age of 20 years, but only 11.4% among those who began drinking at the age of 25 years or above. HED was far more common among regular drinkers than occasional drinkers. Logically, it is impossible for occasional drinkers to engage in frequent HED; hence, the prevalence of frequent HED in this group was 0%. One-fourth of regular drinkers had frequent HED. A high prevalence of HED and frequent HED was observed in drinkers who cited their own or other's home as the most frequent drinking venue. Drinkers with exposure to alcohol advertising had slightly higher HED than those who were not exposed to such advertising. An association between alcohol advertising and frequent HED was not statistically significant. HED and frequent HED were far more common in males. A high prevalence of HED and frequent HED was observed in those who smoked more regularly.

### 3.3 Factors associated with HED and frequent HED

The results of multivariate analysis are shown in Table 3. Drinkers who started drinking at <20 years had higher odds of HED (adjusted OR, 1.43; 95% CI, 1.26–1.62) and frequent HED (adjusted OR, 1.31; 95% CI, 1.12–1.53) than those who started drinking at ≥25 years. Regular drinkers had markedly higher odds of HED than occasional drinkers (adjusted OR, 5.47; 95% CI, 4.98–6.01). Compared to drinking at one's own home, drinking at a party or traditional ceremony was associated with decreased odds of HED and frequent HED, whereas drinking at someone else's home was associated with increased odds of HED but decreased odds of frequent HED. Exposure to alcohol advertising was associated with higher odds of HED but not

**Table 1. Characteristics of current drinkers (n = 23,073).**

| Characteristics | n | % |
|---|---|---|
| Sex | | |
| Male | 18,356 | 79.6 |
| Female | 4,717 | 20.4 |
| Age (years) | | |
| Mean (SD) | 44.3 (14.1) | |
| 15–19 | 734 | 3.2 |
| 20–30 | 3604 | 15.6 |
| 31–45 | 7753 | 33.6 |
| 46–60 | 8095 | 35.1 |
| ≥61 | 2887 | 12.5 |
| Education | | |
| Primary school or lower | 11,635 | 50.5 |
| Secondary school | 9,009 | 39.1 |
| College or higher | 2,391 | 10.4 |
| Household income per month (Thai baht) | | |
| Mean (SD) | 10,919.9 (11,160.3) | |
| <5,000 | 5,440 | 25.0 |
| 5,000–9999 | 7,935 | 36.5 |
| ≥10,000 | 8,381 | 38.5 |
| Marital status | | |
| Single | 4,587 | 19.9 |
| Married | 16,221 | 70.3 |
| Widowed/divorced/separated | 2,261 | 9.8 |
| Smoking | | |
| No | 13,016 | 56.4 |
| Occasional | 1,113 | 4.8 |
| Regular | 8,944 | 38.7 |
| Age at first alcohol use (years) | | |
| Mean (SD) | 20.6 (6.2) | |
| <20 | 11,177 | 48.4 |
| 20–24 | 7,631 | 33.1 |
| ≥25 | 4,265 | 18.5 |
| Frequency of drinking | | |
| Occasional | 13,057 | 56.6 |
| Regular | 10,016 | 43.4 |
| Most frequent drinking venue | | |
| Own home | 9,118 | 39.5 |
| Someone else's home | 5,061 | 21.9 |
| Party or traditional ceremony | 8,894 | 38.6 |
| Exposure to alcohol advertising | | |
| No | 14,053 | 63.6 |
| Yes | 8,053 | 36.4 |
| HED | | |
| No | 18,792 | 81.4 |
| Yes | 4,283 | 18.6 |
| Frequent HED | | |
| No | 20,746 | 89.9 |

(*Continued*)

**Table 1.** (Continued)

| Characteristics | n | % |
|---|---:|---:|
| Yes | 2,327 | 10.1 |

SD, standard deviation; HED, heavy episodic drinking.

frequent HED. Female sex was associated with lower odds of HED and frequent HED. Working age groups had higher odds of HED and frequent HED than teenagers. Drinkers with a college education or higher had lower odds of frequent HED than those with primary school education or lower. Having a monthly income ≥10,000 THB was associated with higher odds of HED and frequent HED.

Table 4 shows the results for each age group from the regression model. Among participants aged 15–19 years, earlier onset of drinking was not significantly associated with HED and frequent HED (Model A). The finding was similar for the 20–24 years age group (Model B). In participants aged 25 years or above, alcohol use onset before the age of 20 years was associated with increased odds of HED and frequent HED (Model C). This is consistent with the result of the full sample model presented in Table 3.

An interaction between age at first alcohol use and sex in terms of their effect on HED and frequent HED was observed, indicating that the effect of age at first alcohol use differed by sex. Table 5 demonstrates that age at first alcohol use had a stronger association with HED and frequent HED in females by the factor of adjusted OR of the interaction term. For example, the adjusted ORs of age at first alcohol use <20 years old were 1.31 and 1.20 for HED and frequent HED, respectively, in males. The corresponding ORs in females were 2.17 (calculated from 1.31*1.66) and 2.44 (calculated from 1.20*2.03), respectively.

## 4. Discussion

One-fifth of Thai drinkers engaged in HED, whereas one-tenth engaged in frequent HED. Lower age at first alcohol use, regular drinking, drinking at home, and exposure to alcohol advertising increased the likelihood of HED. Lower age at first alcohol use and drinking at home were associated with frequent HED. Demographic and socioeconomic factors were also associated with HED and frequent HED.

### 4.1 Association between age at first alcohol use and HED

The association between age at first alcohol use and HED among drinkers in a developing country was investigated using data from the SADBeS 2017. This study demonstrates that, after adjusting for covariates including drinking-related contextual factors (drinking venue and exposure to alcohol advertising), earlier alcohol initiation was associated with engagement in HED and frequent HED among drinkers. Drinkers who started alcohol drinking at <20 years had higher odds of HED and frequent HED, and those who started drinking at 20–24 years had higher odds of HED than drinkers who started alcohol drinking at ≥ 25 years. Nonetheless, this association was not observed when the analysis was limited to the age group of 15–19 years and 20–24 years. This may be due to less variability in drinking behavior in persons from a narrow age range. The analysis of the 25+ age group yielded results similar to those for the full sample model. This finding supports that the association between age at first alcohol use and HED in developing countries is similar to that in developed countries. For example, an analysis of the national survey in the US found that age at first use of alcohol was associated with increased HED in the last 30 days among drinkers [15]. Prospective studies from

**Table 2. Prevalence of HED and frequent HED by drinkers' characteristics (n = 23,073).**

| Characteristic | HED | | | Frequent HED | | |
|---|---|---|---|---|---|---|
| | % | 95% CI | p-value[a] | % | 95% CI | p-value[a] |
| Drinking-related characteristics | | | | | | |
| Age at first alcohol use (years) | | | <0.001 | | | <0.001 |
| ≥25 | 11.4 | 10.5–12.4 | | 6.4 | 5.7–7.2 | |
| 20–24 | 17.4 | 16.5–18.2 | | 9.4 | 8.8–10.1 | |
| <20 | 22.1 | 21.3–22.9 | | 12.0 | 11.4–12.6 | |
| Frequency of drinking | | | <0.001 | | | N/A |
| Occasional | 6.9 | 6.5–7.4 | | 0.0 | 0.0 | |
| Regular | 33.7 | 32.8–34.6 | | 23.2 | 22.4–24.1 | |
| Most frequent drinking venue | | | <0.001 | | | <0.001 |
| Own home | 24.7 | 23.8–25.6 | | 15.6 | 14.9–16.3 | |
| Someone else's home | 22.4 | 21.3–23.6 | | 11.2 | 10.4–12.1 | |
| Party or traditional ceremony | 10.1 | 9.5–10.7 | | 3.8 | 3.4–4.2 | |
| Exposure to alcohol advertising | | | <0.001 | | | 0.083 |
| No | 17.5 | 16.9–18.2 | | 9.6 | 9.2–10.1 | |
| Yes | 19.7 | 18.9–20.6 | | 10.4 | 9.7–11.1 | |
| Demographic and socioeconomic characteristics | | | | | | |
| Sex | | | <0.001 | | | <0.001 |
| Male | 21.5 | 20.9–22.1 | | 11.8 | 11.3–12.3 | |
| Female | 7.3 | 6.6–8.0 | | 3.5 | 3.0–4.0 | |
| Age (years) | | | <0.001 | | | <0.001 |
| 15–19 | 13.2 | 11.0–15.9 | | 4.6 | 3.3–6.4 | |
| 20–30 | 19.4 | 18.1–20.7 | | 9.3 | 8.4–10.3 | |
| 31–45 | 20.1 | 19.2–21.0 | | 10.7 | 10.0–11.4 | |
| 46–60 | 18.3 | 17.5–19.2 | | 10.5 | 9.9–11.2 | |
| ≥61 | 15.3 | 14.0–16.7 | | 9.8 | 8.7–10.9 | |
| Education | | | <0.001 | | | <0.001 |
| Primary school or lower | 18.4 | 17.7–19.1 | | 10.8 | 10.3–11.4 | |
| Secondary school | 19.6 | 18.8–20.5 | | 10.1 | 9.5–10.7 | |
| College or higher | 14.9 | 13.5–16.4 | | 6.3 | 5.4–7.3 | |
| Household income (Thai baht) | | | <0.001 | | | 0.074 |
| <5,000 | 16.6 | 15.6–17.6 | | 9.3 | 8.5–10.1 | |
| 5,000–9999 | 18.4 | 17.6–19.3 | | 10.1 | 9.5–10.8 | |
| ≥10,000 | 19.9 | 19.1–20.8 | | 10.4 | 9.8–11.1 | |
| Marital status | | | <0.001 | | | <0.001 |
| Single | 21.3 | 20.1–22.5 | | 9.9 | 9.1–10.8 | |
| Married | 17.2 | 16.7–17.8 | | 9.6 | 9.2–10.1 | |
| Widowed/divorced/separated | 22.5 | 20.8–24.2 | | 13.6 | 12.3–15.1 | |
| Smoking | | | <0.001 | | | <0.001 |
| No | 13.9 | 13.3–14.5 | | 6.9 | 6.4–7.3 | |
| Occasional | 16.8 | 14.7–19.1 | | 7.1 | 5.7–8.8 | |
| Regular | 25.6 | 24.7–26.5 | | 15.1 | 14.4–15.9 | |

[a] p-value computed using chi-squared test.

HED, heavy episodic drinking.

**Table 3. Factors associated with HED and frequent HED (multivariate logistic regression; n = 20,805).**

| Factor | HED | | | | | Frequent HED | | | | |
|---|---|---|---|---|---|---|---|---|---|---|
| | Crude OR | Adjusted OR | 95% CI | p-value | | Crude OR | Adjusted OR | 95% CI | p-value | |
| | | | | Wald's test | LR test | | | | Wald's test | LR test |
| Drinking-related characteristics | | | | | | | | | | |
| Age at first alcohol use (years) | | | | | <0.001 | | | | | <0.001 |
| ≥25 | 1 | 1 | | | | 1 | 1 | | | |
| 20–24 | 1.59 | 1.24 | 1.09–1.42 | <0.001 | | 1.49 | 1.11 | 0.95–1.31 | 0.189 | |
| <20 | 2.22 | 1.43 | 1.26–1.62 | <0.001 | | 1.97 | 1.31 | 1.12–1.53 | <0.001 | |
| Frequency of drinking | | | | | <0.001 | | | | | N/A |
| Occasional | 1 | 1 | | | | N/A | N/A | N/A | N/A | |
| Regular | 6.70 | 5.47 | 4.98–6.01 | <0.001 | | N/A | N/A | N/A | N/A | |
| Most frequent drinking venue | | | | | <0.001 | | | | | <0.001 |
| Own home | 1 | 1 | | | | 1 | 1 | | | |
| Someone else's home | 0.90 | 1.19 | 1.08–1.31 | <0.001 | | 0.70 | 0.70 | 0.62–0.78 | <0.001 | |
| Party or traditional ceremony | 0.35 | 0.78 | 0.70–0.86 | <0.001 | | 0.22 | 0.26 | 0.23–0.30 | <0.001 | |
| Exposure to alcohol advertising | | | | | <0.001 | | | | | 0.078 |
| No | 1 | | | | | 1 | | | | |
| Yes | 1.14 | 1.16 | 1.07–1.25 | <0.001 | | 1.05 | 1.09 | 0.99–1.21 | 0.078 | |
| Demographic and socioeconomic characteristics | | | | | | | | | | |
| Sex | | | | | <0.001 | | | | | <0.001 |
| Male | 1 | 1 | | | | 1 | 1 | | | |
| Female | 0.28 | 0.55 | 0.47–0.63 | <0.001 | | 0.25 | 0.39 | 0.32–0.47 | <0.001 | |
| Age (years) | | | | | <0.001 | | | | | <0.001 |
| 15–19 | 1 | 1 | | | | 1 | 1 | | | |
| 20–30 | 1.43 | 1.38 | 1.05–1.82 | 0.022 | | 2.07 | 2.26 | 1.48–3.45 | <0.001 | |
| 31–45 | 1.42 | 1.41 | 1.07–1.86 | 0.016 | | 2.31 | 2.64 | 1.73–4.02 | <0.001 | |
| 46–60 | 1.28 | 1.41 | 1.06–1.87 | 0.019 | | 2.27 | 2.78 | 1.81–4.27 | <0.001 | |
| ≥61 | 1.03 | 1.09 | 0.80–1.47 | 0.582 | | 2.10 | 2.40 | 1.54–3.75 | <0.001 | |
| Education | | | | | 0.191 | | | | | 0.010 |
| Primary school or lower | 1 | 1 | | | | 1 | 1 | | | |
| Secondary school | 1.12 | 1.09 | 0.99–1.19 | 0.080 | | 0.96 | 1.02 | 0.91–1.14 | 0.756 | |
| College or higher | 0.76 | 1.02 | 0.87–1.18 | 0.841 | | 0.52 | 0.75 | 0.61–0.93 | 0.007 | |
| Monthly household income (Thai baht) | | | | | <0.001 | | | | | <0.001 |
| <5,000 | 1 | 1 | | | | 1 | 1 | | | |
| 5,000–9999 | 1.16 | 1.01 | 0.91–1.12 | 0.799 | | 1.14 | 1.08 | 0.95–1.23 | 0.257 | |
| ≥10,000 | 1.27 | 1.25 | 1.12–1.40 | <0.001 | | 1.16 | 1.28 | 1.11–1.46 | <0.001 | |
| Marital status | | | | | <0.001 | | | | | <0.001 |
| Single | 1 | 1 | | | | 1 | 1 | | | |
| Married | 0.74 | 0.70 | 0.63–0.78 | <0.001 | | 0.91 | 0.75 | 0.65–0.86 | <0.001 | |
| Widowed/divorced/separated | 1.04 | 1.11 | 0.95–1.30 | 0.172 | | 1.37 | 1.34 | 1.11–1.61 | 0.002 | |
| Smoking | | | | | <0.001 | | | | | <0.001 |
| No | 1 | 1 | | | | 1 | 1 | | | |
| Occasional | 1.26 | 1.01 | 0.84–1.23 | 0.884 | | 1.08 | 0.86 | 0.66–1.12 | 0.257 | |
| Regular | 2.11 | 1.19 | 1.10–1.30 | <0.001 | | 2.46 | 1.65 | 1.49–1.83 | <0.001 | |

LR test, likelihood ratio test; HED, heavy episodic drinking; OR, odds ratio.

**Table 4. Effect of age at first alcohol use on HED and frequent HED in each age group.**

| Model | HED[a] | | | | Frequent HED[b] | | | |
|---|---|---|---|---|---|---|---|---|
| | Adjusted OR | 95% CI | p-value | | Adjusted OR | 95% CI | p-value | |
| | | | Wald's test | LR test | | | Wald's test | LR test |
| Model A: 15–19 years old (n = 547) | | | | 0.399 | | | | 0.146 |
| Continuous | 1.09 | 0.89–1.34 | 0.399 | | 0.81 | 0.61–1.07 | 0.146 | |
| Model B: 20–24 years old (n = 1,197) | | | | 0.093 | | | | 0.305 |
| 20–25 | 1 | | | | 1 | | | |
| <20 | 1.43 | 0.94–2.18 | 0.093 | | 1.33 | 0.76–2.33 | 0.305 | |
| Model C: 25+ years old (n = 19,061) | | | | <0.001 | | | | <0.001 |
| ≥25 | 1 | | | | 1 | | | |
| 20–24 | 1.25 | 1.10–1.42 | <0.001 | | 1.11 | 0.95–1.31 | 0.199 | |
| <20 | 1.42 | 1.25–1.61 | <0.001 | | 1.31 | 1.12–1.53 | <0.001 | |

[a] Adjusted for most frequent drinking venue, exposure to alcohol advertising, sex, age, education, marital status, household income, smoking, and frequency of drinking.

[b] Adjusted for most frequent drinking venue, exposure to alcohol advertising, sex, age, education, marital status, household income, and smoking.

LR test, likelihood ratio test; HED, heavy episodic drinking; OR, odds ratio.

Australia and Canada confirmed this association [10, 12]. Moreover, earlier drinking initiation was associated with alcohol dependence and alcohol use disorder as well as substance use, violence, and employment problems [17–19]. Apart from the direct consequences of alcohol consumption, the initiation of a substance during adolescence may result in future use of other substances [35]. This issue should be examined in individuals with early drinking onset.

Drinking initiation is a result of a combination of contextual factors including parental drinking, parental supervision, friends' drinking, and school bonding [16]. Apart from those who are likely to drink heavily themselves, parents with early onset of drinking perceived providing alcohol to children as acceptable [36]. Hence, an effect of earlier drinking initiation in one generation may lead to HED in the next generation. This underscores the importance of implementing measures to delay drinking onset. A recent study on Chilean youths reported

**Table 5. Interaction between age at first alcohol use and sex (n = 20,805).**

| Factors | HED[a] | | | | Frequent HED[b] | | | |
|---|---|---|---|---|---|---|---|---|
| | Adjusted OR | 95% CI | p-value | | Adjusted OR | 95% CI | p-value | |
| | | | Wald's test | LR test | | | Wald's test | LR test |
| Interaction term (age at first alcohol use and sex) | | | | 0.009 | | | | 0.009 |
| ≥25 and male | 1 | | | | 1 | | | |
| 20–24 and female | 1.43 | 1.02–2.00 | 0.036 | | 1.40 | 0.87–2.25 | 0.164 | |
| <20 and female | 1.66 | 1.19–2.32 | 0.003 | | 2.03 | 1.29–3.20 | 0.002 | |
| Age at first alcohol use (years) | | | | <0.001 | | | | 0.004 |
| ≥25 | 1 | | | | 1 | | | |
| 20–24 | 1.14 | 0.99–1.32 | 0.065 | | 1.03 | 0.87–1.23 | 0.700 | |
| <20 | 1.31 | 1.14–1.50 | <0.001 | | 1.20 | 1.02–1.41 | 0.030 | |

[a] Adjusted for most frequent drinking venue, exposure to alcohol advertising, sex, age, education, marital status, household income, smoking, and frequency of drinking.

[b] Adjusted for most frequent drinking venue, exposure to alcohol advertising, sex, age, education, marital status, household income, and smoking.

LR test, likelihood ratio test; HED, heavy episodic drinking; OR, odds ratio.

the effect of pricing policy on the delayed initiation of alcohol use. The study estimated that 6.6 months delay in alcohol initiation could be brought about by a 10% increase in the price of alcoholic beverages [37]. This warrants the implementation of tax and pricing policies for delaying alcohol initiation and preventing engagement in HED.

The finding of lower odds of HED in females in this study was consistent with the finding from the International Alcohol Control Study, which reported four-fold chance of HED in Thai males than Thai females [30]. This ratio was in the same range as for other developing countries including Mongolia, South Africa, and Brazil [30, 38]. Although females were considerably less likely to engage in HED than males, our interaction model showed that the effect of age at first alcohol use on HED and frequent HED was significantly stronger in females. Hence, tax and pricing policies need to be implemented across the board including alcoholic beverages preferred by young females such as alcopops and cider [39, 40]. The interaction between sex and another determinant of heavy drinking was observed in a study from Brazil: higher education was protective against heavy drinking in males, but increased the chance of heavy drinking in females [38]. Sex likely modifies the effect of several factors on heavy drinking in the context of developing countries. This reflects different underpinning casual models of heavy drinking between males and females.

## 4.2 Effects of other covariates

Our study demonstrated that factors including frequency of drinking, most frequent drinking venue, exposure to alcohol advertising, age, smoking, and marital status are correlated with HED. These factors have been shown to be associated with HED in previous studies on the effects of drinking frequency [22], drinking location [21], alcohol marketing in youths [20, 41], age [21, 42], and smoking [43, 44]. The direction of the associations for most factors is consistent with previous research. The relationship between drinking location and HED, on the other hand, was complicated.

According to a study based on data from a Canadian survey, participants consumed the least amount of alcohol at a restaurant and the most at a bar/disco/nightclub. The amount consumed at home was moderate [45]. In the current study, participants indicating home as the most frequent drinking venue had a higher chance of HED and frequent HED than those mentioning party or traditional ceremony (which includes restaurants). Hence, this seemingly inconsistent finding may be due, in part, to the grouping of drinking locations in our study. Moreover, it should be noted that the drinking venue variable in our study does not directly refer to the location where HED occurred; it refers to the drinking location where participants visited the most for drinking in general.

A study conducted in the United States investigated heavy drinking in both public and private settings. It demonstrated that the other environmental factor, the presence of many intoxicated people at the drinking venue, was linked to heavy drinking in both public and private settings [46]. The current study did not include variables related to the immediate environmental factors of an HED occasion. Further investigation of HED in Thailand should include specific information about each HED occasion to fill this gap.

## 4.3 Limitations

This study has some limitations. Data used in the analysis are from a cross-sectional survey. This may lead to recall bias, especially concerning age at first alcohol use. The survey participants were limited to the Thai-speaking population, whereas some Thai citizens, such as hill tribes in the north, are unable to communicate in Thai. The findings of this study may not be applicable to these populations. The definition of HED used in this study was not the same as

the one used by WHO regarding the time frame of the measurement. The item for HED in the survey used a 12-month time frame, whereas the WHO definition of HED uses a 30-day time frame. The HED variable used in the analysis was created by classifying heavy drinking at least once a month as HED. As a result, the prevalence of HED reported in this study may have been underestimated. Furthermore, the item for HED in the survey was based on the definition of HED for males, whereas the amount of drinking considered HED for females is slightly lower than that of males. As a result, the prevalence of HED in females was also underestimated in this study.

## Acknowledgments

The authors would like to thank the Center for Alcohol Studies, Thailand, for providing access to the data source used in this study.

## Author Contributions

**Conceptualization:** Paithoon Sonthon, Narumon Janma, Udomsak Saengow.

**Data curation:** Udomsak Saengow.

**Formal analysis:** Paithoon Sonthon.

**Funding acquisition:** Udomsak Saengow.

**Methodology:** Paithoon Sonthon, Narumon Janma, Udomsak Saengow.

**Project administration:** Paithoon Sonthon, Narumon Janma.

**Software:** Paithoon Sonthon.

**Supervision:** Udomsak Saengow.

**Writing – original draft:** Paithoon Sonthon, Narumon Janma.

**Writing – review & editing:** Udomsak Saengow.

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
