## [Decision Letter · Decision Letter 0]

14 Jul 2021

PONE-D-21-16712

Association between age at first alcohol use and heavy episodic drinking: An analysis of the Smoking and Alcohol Drinking Behavior Survey 2017

PLOS ONE

Dear Dr. Saengow,

Thank you for submitting your manuscript to PLOS ONE. After careful consideration, we feel that it has merit but does not fully meet PLOS ONE’s publication criteria as it currently stands. Therefore, we invite you to submit a revised version of the manuscript that addresses the points raised during the review process.

We look forward to receiving your revised manuscript.

Kind regards,

Zila M Sanchez, PhD

Academic Editor

PLOS ONE

Journal Requirements:

Reviewers' comments:

Reviewer's Responses to Questions

**Comments to the Author**

1. Is the manuscript technically sound, and do the data support the conclusions?

Reviewer #1: Partly

Reviewer #2: Yes

Reviewer #3: Yes

2. Has the statistical analysis been performed appropriately and rigorously? 

Reviewer #1: Yes

Reviewer #2: Yes

Reviewer #3: Yes

3. Have the authors made all data underlying the findings in their manuscript fully available?

Reviewer #1: No

Reviewer #2: Yes

Reviewer #3: Yes

4. Is the manuscript presented in an intelligible fashion and written in standard English?

Reviewer #1: Yes

Reviewer #2: Yes

Reviewer #3: Yes

5. Review Comments to the Author

Reviewer #1: The authors have undertaken an important subject, that of examining if associations between age of first drink and later HED are observed in a middle-income country (i.e., Thailand), as have been observed in data from several higher-income countries. The dataset is strong, and the topic is important. I have some concerns (some minor, some not) that may help improve the analyses and their contribution to the literature.

Major

1. My major concern is that the analytic sample needs to be revised in order to accurately evaluate cross-sectional associations with age of first drink as defined. The use of the full sample (ages 15 and up) is completely accurate for estimating prevalence of HID and frequent HED among Thai-speakers in Thailand. However, I do not feel it is appropriate when evaluating associations between age at first drink and HED to include cases in the models that have not had the opportunity to be coded fully using the defined age of first drink measure: <20, 20-25, >25. Respondents aged 15-19 have to have an age of first drink at <20; those aged 20-25 do not have the ability to be coded as >25. This skews the analyses and resulting estimates. Since age of first drink was asked as an open-ended measure, I recommend that a series of models possibly be run. One possible approach would be that for respondents aged 15-19, include age at first drink as a continuous measure with controls that are appropriate for the age range (e.g., not including marital status). For respondents aged 20-25, possibly include age at first drink as a dichotomy of <20 vs. 20+. For respondents aged 26+, use the initial trichotomy of <20, 20-25, >25. This would also help address the “telescoping” that happens when attempting to remember age at first drink as one gets older.

2. An additional limitation that should be noted is that only Thai speakers were included; as the population in Thailand includes many hill tribes in various locations (particularly the North), findings may not generalize to these populations.

3. In 2.3 HED, the wording of the item for HED reads, “How often have you drunk heavily (5 drinks)…”. Was this the actual wording used? Were females asked about the prevalence of 4 drinks?

4. No mention of what was done to address missing data on covariates or outcome measures is available. Please state the level of missingness present (or at least a range), and what was done to address this, or if complete case data were used.

5. I find it surprising that a yes/no item could be asked of exposure to alcohol advertising. Were respondents given any further information on what constitutes alcohol advertising?

6. Analyses should have included methods to address the complex sample (i.e., strata, clustering, weights). Please describe.

7. Please provide statistical testing of bivariate associations in Table 1.

Minor

1. The authors sometimes use “developed” and “developing” versus “higher-income” and “middle-income”. Please be consistent.

2. The authors should cite publications from national surveys for their prevalence estimates of nation-specific HID. Citation of a study looking only at HED among the Hispanic population at the US-Mexico border is not really appropriate for national HED prevalence, which can be found in sources such as the National Household Survey on Drug Use and the Monitoring the Future Survey.

3. The data presented on trends in current drinking among Thai adolescents is interesting, but confusing (p. 4). Please clarify.

4. The 2nd to last sentence in the Introduction is not clear.

5. In 2.2 Data source, the text reads 23,070 cases. However, in the sample flowchart, the total is 23,073.

6. On p. 12, I believe the use of the word “attenuated” is incorrect, as this indicates the association was smaller for females than males.

Reviewer #2: The study used national data in Thailand to examine the association between age at first alcohol use and heavy episodic drinking. This is an important issue less focused in research out of the developed countries and is worth exploration. The literature review laid a clear background for the focused topic, and the methods are suitable for the concerning questions and are clearly presented. The discussion interprets the results in the context of existing literature and provides policy implications. Here are some suggestions for the authors’ consideration:

1. The background introduction may be further strengthened by informing readers a little about the social/cultural background in Thailand and how that may contribute to the discrepancy in heavy drinking prevalence between Thailand and developed countries.

2. In the methods section

a. A little more specifics in the introduction of the database and sample. In particular, what is the proportion of the missing?

b. Is the age category of >=25 relative to the other two younger groups too broad?

3. In the result presentation section

a. It is better to provide specific statistics when presenting the results in the texts to facilitate reading

b. Given that this is a cross-sectional study, it is better to say correlates rather than “Determinants of HED and frequent HED”

c. The statement regarding the interaction between gender and initial alcohol use doesn’t seem to match the findings in Table 4 and is in contrast with its following statement: “however, female sex attenuated the effect of age at first alcohol use on HED and frequent HED. For example, in females, the effect of age at first alcohol use <20 years on HED and frequent HED was increased by 66% and 103%, respectively. ” Female earlier onset seems to increase the risk of HED instead.

4. The discussion can be further enriched by more closely relating the findings to relevant literature. For example, evidence concerning gateway theory in substance use in general may help strengthen the discussion from early drinking onset to HED (e.g., Zhang, S., Wu, S., Wu, Q., Durkin, D. W., & Marsiglia, F. F. (2021). Adolescent drug use initiation and transition into other drugs: A retrospective longitudinal examination across race/ethnicity. Addictive Behaviors, 113, 106679.)

Reviewer #3: Major comments

The study is relevant for bringing results about HED in amiddle-income Asian country.

The results are interesting and in line withprevious literature on the subject.

Outcome variables have limitations that arewell described.

As contributions to the improvement of the report, I suggest:

1 - Review the use of the term gender, as apparently thevariable used was sex. For this, please consider the SAGER guidelines (https://www.equator-network.org/reporting-guidelines/sager-guidelines/)

2 - Emphasize, in the discussion, the main differences in theresults observed in Thailand in relation to other countries, in order toreinforce the importance of carrying out the study. Also inform about public policies to combat the harmful use of alcohol in force in Thailand, and how the results of the study can contribute to the improvement of such policies. 

Minor comments and suggestions

Title:

- The title should include the name of the country where the survey was conducted.

Abstract:

- Please, use the same objective, as stated at the end ofthe introduction: “This study aimed to investigate the association between ageat first alcohol use and HED using data from a nationally representative surveyin Thailand.”

- The abstract should show effect sizes and confidence intervals. “Regular drinking, drinking at home, and exposure to alcohol advertising increased the likelihood of HED.” Also, it is suggested to replace “likelihood”by “chance”, which is more adequate to the type of analysis used (logisticregression).

- “There was a significant interaction between the effect ofage at first alcohol use and gender on HED and frequent HED.” Please, indicate the direction of the associations, eg: lower or higher age? Male or female?

- Please, review the use of the word “gender”.

- “This study provides evidence from middle-income countries that early onset of alcohol use is associated with HED.” – a middle income country

Keywords:

- Please include a keyword related to study design(cross-sectional studies)

Introduction:

- It should include data on HED prevalence in Thailand. Data on theburden of disease caused by alcohol could also be included.

Methods:

- At the “study design” subsection, please indicate clearly the study design (cross-sectional)

- Please, review the use of the word “gender”.

- Please, explain how “exposure to alcohol advertising” was measured and what was considered as “alcohol advertising”.

- Statistical analyses should include the criteria used for including and maintaining variables in the adjusted regression model.

- Ethics: please, indicate how participants gave consent.

Results:

- The results are clearly presented.

- The higher prevalence of HED while drinking at home couldbe emphasised in the text.

Tables 3 and 4: please, indicate as footnote the name of thestatistical tests used to calculate the p-values.

Discussion:

At the begining of this section, sex differences could also be mentioned. Also, sex differences could be more discussed. The discussion could be deepened and include dialogue with studies carried out with other middle-income countries, including Asian countries, and countries in other continents, such as Brazil, Mexico, South Africa.

6. PLOS authors have the option to publish the peer review history of their article (what does this mean?). If published, this will include your full peer review and any attached files.

Reviewer #1: No

Reviewer #2: **Yes: **Saijun Zhang

Reviewer #3: No

---

## [Author Response · Author response to Decision Letter 0]

28 Aug 2021

The response has been uploaded as a separate file.

---

## [Editor Report · Decision Letter 1]

14 Sep 2021

PONE-D-21-16712R1Association between age at first alcohol use and heavy episodic drinking: An analysis of the Thailand's Smoking and Alcohol Drinking Behavior Survey 2017PLOS ONE

Dear Dr. Saengow,

Thank you for submitting your manuscript to PLOS ONE. After careful consideration, we feel that it has merit but does not fully meet PLOS ONE’s publication criteria as it currently stands. Therefore, we invite you to submit a revised version of the manuscript that addresses the points raised during the review process.

We look forward to receiving your revised manuscript.

Kind regards,

Zila M Sanchez, PhD

Academic Editor

PLOS ONE

Journal Requirements:

Additional Editor Comments:

The authors had answered most of the reviewers suggestions.

However some aspects should be revised:

- English edit of the manuscript

- the p values on the tables are sometimes not well placed. A p value for each category of answer is expected when when you have a variable with 3 or more categories (for example: age: 15-19, 20-24 and 25+ - here you should have 2 p values: 15-19 is the reference and 20-24 has one p value and 25+ has another p value). Most of the tables presents at least one error on the report of the p values).

- include in the methods section that missing data was excluded and that the analysis refers to complete cases only.
---

## [Author Response · Author response to Decision Letter 1]

30 Sep 2021

Please see the response to reviewer file upload along with the manuscript file.

---

## [Editor Report · Decision Letter 2]

22 Oct 2021

Association between age at first alcohol use and heavy episodic drinking: An analysis of Thailand's Smoking and Alcohol Drinking Behavior Survey 2017

PONE-D-21-16712R2

Dear Dr. Saengow,

We’re pleased to inform you that your manuscript has been judged scientifically suitable for publication and will be formally accepted for publication once it meets all outstanding technical requirements.

Kind regards,

Zila M Sanchez, PhD

Academic Editor

PLOS ONE
---

## [Editor Report · Acceptance letter]

28 Oct 2021

PONE-D-21-16712R2 

Association between age at first alcohol use and heavy episodic drinking: An analysis of Thailand’s Smoking and Alcohol Drinking Behavior Survey 2017 

Dear Dr. Saengow:

I'm pleased to inform you that your manuscript has been deemed suitable for publication in PLOS ONE. Congratulations! Your manuscript is now with our production department. 

Kind regards, 

on behalf of

Dr. Zila M Sanchez 

Academic Editor

PLOS ONE